# Explaining the UK′s 'high-risk' approach to type 2 diabetes prevention: findings from a qualitative interview study with policy-makers in England

Eleanor Barry ⬥ , Trisha Greenhalgh ⬥ , Sara Shaw ⬥ , Chrysanthi Papoutsi

Nuffield Department of Primary Care Health Sciences, University of Oxford, Oxford, UK

**Correspondence to**
Dr Eleanor Barry;
Eleanor.barry@phc.ox.ac.uk

## ABSTRACT

**Objectives** When seeking to prevent type 2 diabetes, a balance must be struck between individual approaches (focusing on people's behaviour 'choices') and population approaches (focusing on the environment in which those choices are made) to address the socioeconomic complexity of diabetes development. We sought to explore how this balance is negotiated in the accounts of policy-makers developing and enacting diabetes prevention policy.

**Methods** Twelve semistructured interviews were undertaken with nine UK policy-makers between 2018–2021. We explored their perspectives on disease prevention strategies and what influenced policy decision-making. Interviews were transcribed and analysed thematically using NVIVO. We used Shiffman's political priority framework to theorise why some diabetes prevention policy approaches gather political support while others do not.

**Results** The distribution of power and funding among relevant actors, and the way they exerted their power determined the dominant approach in diabetes prevention policy. As a result of this distribution, policy-makers framed their accounts of diabetes prevention policies in terms of individual behaviour change, monitoring personal quantitative markers but with limited ability to effect population-level approaches. Such an approach aligns with the current prevailing neoliberal political context, which focuses on individual lifestyle choices to prevent disease rather than on infrastructure measures to improve the environments and contexts within which those choices are made.

**Conclusion** Within new local and national policy structures, there is an opportunity for collaborative working among the National Health Service, local governments and public health teams to balance the focus on disease prevention, addressing upstream drivers of ill health as well as targeting individuals with the highest risk of diabetes.

## INTRODUCTION

Type 2 diabetes is an enduring cause of morbidity and mortality. As its prevalence continues to increase, prevention has become a global health priority.[1] Geoffrey Rose outlined two approaches to disease

---

### STRENGTHS AND LIMITATIONS OF THIS STUDY

⇒ This is the first analysis of diabetes prevention policies in the UK, using a political science perspective.

⇒ We have explored the policy process as well as the research–policy gap using Shiffman's political priorities framework.

⇒ However, our focus on a small number of elite interviews means that our findings are preliminary.

---

prevention. The first, 'whole-population' approach, introduces societal level changes to reduce everyone's risk of disease by a small amount. The second, 'high-risk' approach, identifies individuals with disease risk factors and focuses on individual interventions (medication or lifestyle programmes) to reduce disease incidence. There is a continuum of disease risk and reducing the population's risk by a small amount is thought to be more effective than focusing on high-risk individuals. Despite this, a number of western countries (eg, USA,[2] Australia[3] and New Zealand[4]) have chosen to adopt mainly or exclusively a high-risk approach, identifying those considered at increased risk and offering them lifestyle education programmes. In England, the National Health Service (NHS) National Diabetes Prevention Programme was introduced in 2016[5] and from April 2021, General Practitioners (GPs) were incentivised to follow NICE (National Institute for Health and Care Excellence) guideline PH (public health) 38,[6] to identify and maintain a register of those with 'non-diabetic hyperglycaemia' ('pre-diabetes'), and in some areas offer annual reviews and referrals to lifestyle interventions. In the UK, diabetes prevention sits within the NHS with local public health teams commissioned to support policy implementation or evaluations.

The NHS National Diabetes Prevention Programme consists of 13 classroom-based

lifestyle intervention sessions delivered over 9 months,[5] supported by trial evidence showing reductions in the risk of disease if individuals engage and complete the interventions.[7 8] The programme has helped individuals to improve their weight and glycated haemoglobin (HbA1c). However, evaluations have highlighted several concerns.[9] First, there were high attrition rates from initial referrals to programme completion, with only 19%–22% of those referred completing the programme.[9 10] This is likely due to how individuals are informed, internalise and contextualise the pre-diabetes diagnosis.[11] Second, there are reported concerns over variations in the intervention's accessibility, quality and fidelity.[12–18] Evaluations have also identified limited improvements in weight and glycaemic markers in women, black and Asian ethnic groups and people from lower socioeconomic backgrounds.[9 19] Weight loss is difficult, especially for those who face structural barriers.[20 21] For instance, the poorest 10% of the country would need to spend 75% of their disposable income to meet the NHS's Eatwell guidelines.[20] Hence, while current policies help some individuals reduce their risk, there is a concurrent risk of widening health inequalities if these engagement patterns and outcomes continue.

The UK government's population-based approach to tackling rising obesity levels (and subsequently non-communicable diseases such as diabetes and cardiovascular disease) was updated during the COVID-19 pandemic.[22] The strategy currently consists of a soft drinks levy, a school-based approach to sugar reduction, increasing physical activity and food calorie labelling initiatives. These population-level initiatives have been criticised by diabetes organisations and public health leaders as inadequate to reduce the burden of long-term conditions and address health inequalities because they do not tackle the underlying drivers of ill health.[20 21 23–27] Plans to restrict supermarket two for one promotions on foods high in fat, salt and sugar, as well TV advertising restrictions on unhealthy foods before 21:00 have been delayed, a decision described by some as an 'act of supreme self-harm'.[22 26 28]

In England, public health teams within local governments are tasked with tackling the community-level determinants of health. Public health teams transitioned to the local authority setting 10 years ago and have had to build new relationships, logics and ways of working to establish themselves within a different context. Sustained public health budget reductions (up to 25%) have undermined action on a place-based approach to disease prevention.[23] The result is huge variations in local authorities' commitments to improving population health[23 29] and impacts on the working relationships across local authorities and the NHS.[30]

Health behaviours contribute to the development of non-communicable disease such as diabetes. However, it is the social determinants of health, which lead to these behaviours and subsequent health inequalities.[7 20 23] Population health strategies to reduce diseases such as diabetes are limited and poorly financed.

## Aims

We explore how policy decisions are made in implementing national diabetes prevention directives in local settings and identify key influences that shape, enable and constrain these decisions.

## Research questions

In relation to England's policy on type 2 diabetes prevention:
1. What do policy-makers describe as key influences on decision-making for diabetes prevention?
2. Why is England pursuing a largely 'high-risk' strategy to diabetes prevention strategy?

## METHODS
### Study design

We undertook a qualitative semistructured interview study with a maximum variation sample of policy stakeholders involved in developing and implementing diabetes prevention policy.

### Sample and setting

We sought perspectives from a range of local and national policy-makers, using a combination of purposive and snowball sampling. We first selected four policy-makers with experience of local or national policy-making, ensuring representation of public health and NHS settings plus experience in commissioning, health consultancy, clinical practice and public health. We then recruited a further nine stakeholders via participant recommendations.

### Data collection

Nine semistructured interviews with ten stakeholders took place in-person between 2018 and 2020 (one interview was done jointly with two stakeholders). Three stakeholders were then reinterviewed (online via Microsoft Teams due to COVID-19 restrictions) in 2021 to ascertain how the COVID-19 pandemic influenced decision-making (see table 1). Each interview lasted 30–90 min, using preprepared topic guides (see online supplemental file 1 for an example). Topic guides were customised iteratively for each interview, based on the information gathered from previous interviews. We adopted a semistructured approach, enabling the interviewees to express and discuss influences on the policy process from their perspectives. Prompt questions served to elicit how research evidence was used practically to guide decisions. Interviews were audio-recorded (with written consent), anonymised and then transcribed verbatim by a professional transcriber. Each audio recording was checked against each transcription to ensure accuracy by EB.

Interviewees at local authority level included six policy-makers with a public health background, two commissioners, one strategist, a public health consultant and a director of public health. A public health policy-maker working at a national level was also interviewed. NHS participants included a GP clinical lead, a GP primary care network clinical director, a primary care commissioner

**Table 1** Study participants

| Stakeholder interview | Role | Interview date | Follow-up | Sex |
|---|---|---|---|---|
| 1 | PH strategist | 08 August 2018 | | F |
| 2 | PH commissioners 1 and 2 | 06 September 2018 | | 2 M |
| 3 | GP diabetes led | 26 September 2018 | 04 June 2021 | F |
| 4 | Primary care commissioner | 03 October 2018 | | F |
| 5 | GP and PCN clinical director | 07 October 2019 | 27 April 2021 | M |
| 6 | PH national policy-maker | 21 October 2019 | 16 June 2021 | M |
| 7 | PH consultant | 10 December 2019 | | F |
| 8 | DPH | 17 Janauary 2020 | | F |
| 9 | NHSE policy-maker | 18 May 2021 | | F |

In order to maintain participant anonymity, it is not possible to give further details on their roles

PH (Public Health), GP (General Practitioner), PCN (Primary Care Network), DPH (Director of Public Health), NHSE (NHS England)

and former national healthcare policy-maker. The two GP leads and the national public health policy-maker were reinterviewed in 2021 to ascertain if the policy process had changed over the course of the COVID-19 pandemic and to gather feedback on initial findings. Their feedback was that COVID-19 had amplified existing issues, rather than highlighting anything new. All interviewees were approached by email and none of the participants refused to take part. EB conducted the interviews, took field notes and maintained a reflective journal which assisted in the data analysis. This study formed part of a doctoral research study undertaken by EB. TG, SS and CP assisted in the interpretation of the data as well as paper revisions.

## Data analysis

The corresponding author (EB) first used line to line coding with an initial set of three interviews, allowing us to develop a coding framework. This was refined as we coded the remaining interviews, informed by reflections and memos made during the interviews and analysis. We then grouped codes under preliminary themes describing the most significant influences on the policy process, that is, decision-making about policy taking place in a complex system influenced by different structural factors and socioeconomic contexts.[31] Finally, we used Shiffman's framework to extend analysis of how national diabetes prevention policies are interpreted and put into practice locally. NVivo V.12 was used to manage the data.

## Theoretical framework

Originally developed to investigate the determinants of global priorities for health, we used Shiffman's framework[32] to explore how national directives on diabetes prevention are interpreted and implemented at a local level and understand why disease prevention policies focus on targeting high-risk individuals rather than upstream population-level interventions. The framework consists of four domains. For the first domain of 'actor power',

we combined this domain with Lukes three dimensions of power, to give us an additional understanding of how power operates. Lukes understood power to be a multidimensional social influence,[33 34] which allowed us to explore how individuals and organisations use their power in the policy process and how this influences decision-making. For the second domain, 'political context', we unpacked the influence of a neoliberal political system on health policy. For the third domain, we explored the power of ideas behind the policy approach, in particular the emphasis on individual responsibility. Finally, for the fourth domain of 'issue characteristics', we explored what aspects of pre-diabetes make it amenable to a high-risk disease prevention policy.

## Researcher perspective

Adopting an interpretivist approach, we sought to understand how prevention policies are socially and culturally shaped while exploring the complexities underpinning the policy process. As is usual in qualitative research, the study findings were influenced by the context in which data was gathered and assumptions of the researchers.[35] We acknowledge that our identities and professional roles (as clinicians and academics in social sciences, policy and public health) are likely to have influenced data collection and analysis.[36] A GP carried out the interviews, which may have influenced how people within the interviews volunteered and shaped their answers.

The Consolidated criteria for Reporting Qualitative research checklist was used in reporting this qualitative study.[37]

## Patient and public involvement

No patients were involved in this study.

## RESULTS

Below, we set out our findings on what influences decision-making in the translation of national diabetes prevention

directives to local settings. Within the interviews, we explored with policy-makers the policy process and the role of research evidence in the decision-making process. Overall, we found that disease prevention was discussed on an individual basis, focusing on preventing diabetes in those labelled with 'pre-diabetes' with research evidence used in a limited way to support predetermined perspectives. We have structured our findings according to the four domains of Shiffman's framework exploring why this dominant policy approach currently exists.

### Power and funding shaping individualist policy
#### Direct power
In this section, we will explore the use of different forms of power in the policy process and how it is used by actors. The use of direct power by NHS England was identified by all stakeholders as a key influence in diabetes prevention policies. The NHS is currently taking a 'high-risk' approach to diabetes prevention, targeting people labelled as being at high risk of developing disease to change their lifestyles. This is being commissioned via the NHS GP contract and the NHS Diabetes Prevention Programme. Stakeholders reported that money and power sit within the NHS, whose scope is to focus on individual disease development, therefore an individualist discourse will likely be the overarching policy approach (quote 1 in box 1).

Our interviewees gave examples of how NHS England exercised direct influence on the policy process. Providers contracted to deliver the NHS Diabetes Prevention Programme were commissioned directly by NHS England, with reports of limited input from local NHS commissioners. Information and funding were reported to flow in a top–down fashion, with very little scope for local policy-makers to feed into the decision-making processes. Resistance or feedback from local policy-makers seemed to have limited influence on outcomes (quote 2 in box 1).

#### Local influence, but without authority
Comparatively, funding to public health and other government departments has significantly reduced. As a result, public health stakeholders talked about their role as having 'influence without authority' and 'getting stuff done with other people's money' (SH7 Public Health Consultant). Two strategies were used to exert their influence. First, they operationalised power from their knowledge and expertise. Tools used included collecting resident narratives to influence political decision-making. The 'lived experience' and story imagery engaged politicians who understood individual narratives from working in their constituent surgeries, illustrated by quote 3 in box 1.

Second, public health stakeholders used working relationships and trust building to exert their influence, see quote 4, box 1. In the transition from the NHS to a local authority setting, public health teams discussed working to create new identities and working relationships within

---

**Box 1  Money and power shaping individualist policy**

1. 'So, using diabetes as an example, because it seems to be very stuck in the biomedical model which means that the way it's going, we're only really going to get DPP as the solution to preventing diabetes. Because all the money's gone to NHS England, so you can talk to any of the other government departments, they say NHS, the NHS has taken all our money.' (SH6 National Public Health Policy Official)

2. 'We hadn't been informed that the (project) extension had been further extended by additional four months, until the MOU [memorandum of understanding] was sent through to us. And that additional four months didn't reflect the amount of mobilisation fund that we received from NHS England. And our targets for initial assessments had been increased. There was very little leeway in terms of conditions. Which we then tried to (address) but didn't get far with that.' (SH1 Public Health Strategist)

3. 'But they [politicians] are also very much driven by, you know, being part of the communities that they represent, the stuff that walks through their, you know, their weekly surgeries about the challenges that residents are facing and so they will be informed by that kind of sense of anecdote and story.' (SH8 Director of Public Health)

4. 'We've got to kind of build those relationships. We've got to understand, get out and about understanding Local Government to start with because I think, you know, clearly when we joined, we had very little understanding generally probably of what Local Government is and does and how it works so it has been that kind of journey of understanding and building relationships, building trust and only through that are you going to, ever achieve anything really I think.' (SH8 Director of Public Health)

5. 'A conversation last week about PCNs, it's all very well us coming in with ambitious big population health stuff and they're, they're so underfunded and new and developing what they're going to be primarily occupied with is, 'We've just received this national DES (directly enhanced service – that is, funding for a specific initiative) specification and we've got to work out how we've got to go and deliver it, how can you help us do that?' That's the kind of normal conversation.' (SH8 Director of Public Health)

6. 'I sat in a room with a finance man when I was trying to come up with my business case. I had four business cases at one point, on diabetes. And I said to him, 'if you're diabetic and you're type one, you live twenty years less than somebody - if you're type two, you live ten years less, if you have -', you know?' (SH4 NHS Commissioner)

---

local government. Key to this relationship building was understanding each stakeholder's perspectives, priorities and recognising funding imbalances. Public health policy-makers discussed needing to set aside ambitious agendas in to maintain working relationships. For example, quote 5 in box 1 illustrates how engagement with NHS partners was confined to restricted parameters to achieve operational goals.

### Limited power of research evidence
Academic research was discussed as having limited influence on decision-making in the policy process. Interviewees discussed (what they saw as) researchers' lack of understanding of the policy-making context and the complexity of the policy process. Public health

> **Box 2    How political context shapes diabetes prevention**
>
> 1. 'Yeah, so we tried to have that probably a bit ahead of our time a few years ago and absolutely given the kind of financial circumstances and any concern about doing anything about that would lessen local government revenue we didn't win out…' (SH8 Director of Public Health)
> 2. 'Yeah, so you get stories where it is politically, one has to phrase things in Tory areas differently. So, there's a really nice session I went to as part of training in my last job where democratic services shared how they'd gone from a Labour administration to a Tory administration and they kept on the same, they went through the manifestos with a fine-tooth comb. They made lots of notes and then they renamed all their programmes. They kept all their programmes; they kept all their funding, but they renamed them to fit with the ethos of the incoming administration so the holiday hunger programme which makes perfect sense to me, holiday hunger became social eating because it's just nicer and so there's certain things that, that conservative authorities will make all the right decisions and all the right thoughts and feelings. It just needs to be dressed up slightly differently so it's not, it's never a class war thing. It's a sort of nice to do or better if thing.' (Public Health Consultant SH7)
> 3. 'Academic research doesn't recognise the context in which that then has to get operationalised. I suppose the gap in translational public health… I do think maybe there is something about face time in local government I think for academics. I think they just don't understand local government what it is and does and how it works genuinely, and I think there's, a lack of social evidence'. (Director of Public Health SH8)

> **Box 3    Framing ideas about disease prevention as individual responsibility**
>
> 1. 'How can we help that person? How can they be empowered? And so that they are self-caring, and it's their - it's their lives, rather than this you know, this deferring all the time to the doctor… it's about trying to get people to, to have that ownership of their own health, and the fact that they can make a change… whose intervention, is it? So, responsibility is not on us to provide, provide you with a tablet. Our responsibility is to provide you with education and support.' (SH3 NHS Commissioner)
> 2. 'The things I found frustrating; 1) everything's individual, 2) the level at which we put in interventions, I think it's homeopathic. You've got whole population, you need something at scale and then we have like, 'Well we've got a cooking project here and we've got our sort of reading after school project here,' and the NHS commissioning our local diabetes prevention project… we've got 8000 pre-diabetics but provide a service for a quarter of them… (GPs) drive up their (commissioned) activity and then box checked and result, but that doesn't actually work for diabetes prevention.' (SH7 Public Health Consultant)

stakeholders thought that research and guidelines did not reflect the sociopolitical complexities they were operating in, and that research was still orientated to an NHS biomedical perspective (see quote 3 in box 2).

Research evidence was at times used selectively to fit with the policy-makers particular need. For example, to question current practices, argue for their revision or to support the development of pilot schemes. Examples were given where research headlines were used to strengthen the rhetoric in funding applications, as illustrated by quote 6 in box 1 from an NHS commissioner.

### How political context shapes diabetes prevention

Policy-making occurs within a political environment in which actors operate, influencing how we perceive health and illness and what policies are politically acceptable. England might be said to have a neoliberal political system based on ideals of competition, free markets with minimal state intervention and corporate regulation to promote economic prosperity.[38–41] Neoliberal systems reject the idea that health is the product of structural forces placed on individuals, diminishing the role of social, economic and commercial influences on health.[42] In this system, governments might be reluctant to use population-level structural interventions to address upstream drivers of ill health.[38 39 43 44] Instead downstream individual level lifestyle interventions, focused on a narrative of individual responsibility are used as disease prevention strategies.[40 43 45]

We found examples of neoliberal influences on policy-making within the interviews. Working within the local authority setting, public health initiatives were discussed as needing to align with political viewpoints and ideas. There were examples of public health policies, targeting upstream influences on health (at a community level), not gathering political support because of their impact on local economic revenues, see quote 1 in box 2.

Policy-makers discussed trying to align public health strategies with political agendas, to ensure continued funding for community-level interventions. This involved tailoring their language and messaging to fit the local administrations political ethos. Quote 2 in box 2, given by a public health consultant, shows how the language around initiatives and interventions were adapted to remain politically neutral.

### Framing ideas about disease prevention as individual responsibility

In this section, we explore how policy-makers framed ideas around disease prevention, particularly in terms of individual responsibility. Policy-makers working within the NHS believed diabetes could be prevented by empowering individuals through increasing their knowledge via interventions. This emphasis on empowerment and education was assumed to lead to less reliance on the medical system, see quote 1 in box 3, from an NHS commissioner.

Policy-makers working in local authorities and public health framed disease prevention in a different way talking more about upstream community-level influences on health. However, they expressed frustration that the emphasis and funding for disease prevention was largely situated within the NHS. NHS structures such as the GP contract and primary care commissioning cycles were

---

> **Box 4  Characteristics of diagnosing, managing and monitoring pre-diabetes**
>
> 1. 'Like things like grip tests, and step tests, and stuff like that - if we're getting improvements after six weeks, then they are indicative of improvements over a longer period of time. So, it's - We're not going to see a reduction in diabetes, but if a hundred and fifty people finish a course, and after six months they've maintained their behaviour, and we've seen a ten to twenty percent improvement in grip strength, or their ability for their heart to come back to a normal rate after a step test, that's indicative of a health improvement.' (SH2 Public Health Commissioner)
> 2. 'Whether they sustain the physical activity is quite, you know, helpful to know. So even though it's - We're not, we're not measuring you know, a diabetes measure, but if they could still say to you 'yes, I went to this and I'm still exercising twice a week' in ten years' time, then that intervention was worthwhile, regardless. Because it's made them behave - you know, they weren't doing it before, they've been done an intervention and they've sustained that for ten years. Surely that's a positive outcome?' (SH3 GP)
> 3. 'Because I think it's important to sort of work out why people choose, make the choices that they make in a shop, at that time. You're here, what are you having? How did you, how did you make that? What made you come into the shop in the first place? You know? Are you aware of your, of your risk? Are you aware of how they cook this?' (SH3 GP)

seen to perpetuate this individualist discourse due to the 'homeopathic' individual, small scale commissioned interventions and how they were evaluated by surrogate markers (see quote 2 in box 3).

Similarly, most research in diabetes prevention was discussed as being undertaken on an individual level through intervention trials. This is in part due to diabetes being framed as a failing of individual biology and measured via outcomes such as weight and HbA1c in trials and in diabetes prevention reviews. This aligns with the views of policy-makers that diabetes can be prevented by increasing an individual's knowledge and monitoring them through the process, as discussed previously.

### The characteristics of diagnosing, managing and monitoring pre-diabetes

In this section, we explore the features of pre-diabetes diagnosis and management, which construct the individual-level disease prevention strategy (as discussed by policy-makers). First, risk factors for developing diabetes can be identified and quantified in a primary care setting, making it possible for GPs to monitor individuals and detect diabetes development. Because the diagnosis is made via a blood test and in turn interpreted and communicated via a GP, it categorises the condition as a biomedical risk. Second, initial lifestyle advice is given in primary care settings, medicalising individual lifestyles.

From a commissioning and contracting perspective, it is possible to use patient-level surrogate markers (from GPs or commissioned interventions) to monitor targets and use these as terms for payment. Our interviewees emphasised and valued the collection of quantitative data as a way to monitor GP activity. This data was recorded and reflected back to practices in the form of dashboards to compare practices with each other. This data collection process allows the treatment and management of pre-diabetes to fit into short-term funding and commissioning cycles. Quote 1 in box 4, from a public health commissioner, illustrates how small improvements in downstream markers were used as evidence for intervention adherence and success.

Similarly, how pre-diabetes is defined and measured clinically lends itself to being researched in a similar way using quantitative methods such as the analysis of GP data sets, or lifestyle intervention trials. These largely support an individualist approach to reducing diabetes by targeting individuals at high risk.[7 46 47] Although this research did help when making the case for funding initiatives (see box 1 in quote 6), policy-makers reported that this type of research did not reflect the real lives of their patients or explain why they make their life choices, identifying a need for applied social science research (see box 4 in quotes 2 and 3).

## DISCUSSION
### Summary of findings

Our findings show that the UK is largely employing what Geoffrey Rose described as individual-level high-risk disease prevention policies. Informed by the work of Shiffman and Lukes, we identified three dimensions of power and how this is operationalised to shape such policy.[48 49] Power is exerted: (a) directly via top–down NHS England policy directives such as the NHS Diabetes Prevention Programme, (b) indirectly with public health professionals using their expertise to subtly influence the policy agenda and (c) through our neoliberal political context shaping how people think about disease prevention. In the interviews, diabetes prevention policy was framed in terms of individual responsibility to reduce the risk of disease. This is amplified by measurable downstream characteristics used in policies and interventions, such as weight and HbA1c. In addition, lifestyle advice given in primary care medicalises socially constructed lifestyles and neglects the complexity of how lifestyles develop over times.[46 50]

COVID-19 disproportionately affected those with diabetes and obesity. This led to calls to address the social determinants of health[27 51–54] and tighter regulations on the food industry.[55] However, our analysis of how our political context shapes disease prevention policies, suggests that a whole-population approach to diabetes prevention may be politically problematic. National infrastructure changes (such as limiting fast food outlets or restricted advertising) goes against the neoliberal ideals of a 'free

market' economy and involves regulating profit-making corporations.[56] As illustrated by our interviews, reducing local or national economic revenues was deemed to be politically unfavourable.

## How health policies drift from the social determinants of health to individual solutions

Health policies acknowledge the central role of the social determinants of health in non-communicable disease development. However, analysis of the UK government's obesity strategy illustrates that solutions to these problems tend to be based on individuals minimising their health risks by becoming responsible, self-governing citizens, changing their behaviours rather than tackling upstream influences on health.[28 38–40 43 45] Two mechanisms of this 'lifestyle drift' phenomenon were discussed in the interviews.[40] The first mechanism was moving public health teams into local authorities. This placed the responsibility of disease prevention into smaller communities away from national bodies. Focusing on communities to prevent disease puts a focus on individuals within communities to become responsible self-governing citizens, changing their behaviours for the wider benefit of the community.[40] This has been reinforced by the removal of Public Health England (PHE) as a national body leaving a potential national leadership void for local public health teams. A second example of lifestyle drift demonstrated in this study is the positioning of disease prevention in primary care. This medicalises disease risk and lifestyles, further emphasising the need for individuals to use their agency to prevent diabetes. This detracts attention away from the socioeconomic causes of disease (such as the obesogenic environment)[45 56 57] and oversimplifies the complexity of diabetes development.[21] Inherent in this strategy is the assumption that health inequalities are due to individual life choices, not the underlying social causes of disease.[45 56] Subsequent individual health promotion messaging in lifestyle interventions promotes the belief that people should be able to overcome their structural barriers, making rational choices despite difficult living circumstances.[45 56] Framing the responsibility of disease prevention in this way is believed by some academics to be an incredibly powerful industry tactic to deflect any responsibility for disease development, with the targeting of upstream influences (such as removing sugar subsidies) framed as excessive government involvement in people's everyday lives (the nanny state).[58 59]

## Implications for policy and practice

Our findings identified an unequal distribution of power and funding between the NHS and Public Health with regard to prevention policy. This study has shown that the scope of the NHS largely focuses on the individual. Two major health reforms are currently underway in England. CCGs (Clinical Commissioning Groups) are being transformed into integrated care systems (ICSs) with general practices organised into Primary Care Networks tasked with improving community health, prioritising disease prevention and reducing health inequalities.[60] For the ICSs to fulfil their population health portfolios, it is vital they form partnerships and effective working relationships with local authorities and public health teams.[61] There have been calls for greater partnerships and collaborations between the two communities for several years.[62–64] The President of the Faculty of Public Health and Chair of the Royal College of General Practitioners recently published an editorial calling for meaningful collaboration and integration between the two specialities, despite examples of excellent partnerships, these are limited.[65] PHE has also been reorganised, with its health improvement functions moving to the Department of Health and Social Care with a new Office of Health Promotion and Disparities under the leadership of the CMO.[66] This may leave a void of national leadership advocating for reducing health inequalities and addressing the social determinants of health, with reduced support for local public health teams. There will always be a need for individual-level interventions based in primary care; however, to reduce the prevalence of conditions such as diabetes, these must be done in parallel with upstream population-level initiatives addressing the wider determinants of health.[21 26 27 51 67]

## Implications for research

This analysis has illustrated several research–policy gaps in disease prevention. Policy-makers did not feel that research considered their own sociopolitical contexts, emphasising that policy development was a messy, non-linear process taking place within a sociopolitical economic context. These views are supported by senior policy-makers who have written on the limitations of academic evidence.[68] The stakeholders discussed a need for increased partnerships between policy and academia to understand each other's needs, world views and operational constraints. Working together to coproduce from the outset of research to disseminating findings may help bridge translational research gaps. This approach aligns with Weiss's interactive model of research utilisation,[9] but requires a shift away from the traditional hierarchy of biomedical research[69] with a greater focus on researching the social determinants of health instead of downstream individual-level interventions.[70 71]

## Comparison to the literature

While others have already examined the limitations of western health promotion strategies,[42 72] the evidence–policy gap[69 73] and the limitations of evidence-based medicine informing policy,[74] this is the first qualitative study undertaken to examine the policy process of operationalising a national directive into a local initiative from a policy-makers perspective, as well as trying to explain why individual-level policies dominate the policy agenda.

Our study resonates with the findings of other social and political scientists. Baum *et al* undertook a policy analysis examining why individual-level health promotion policies continue to be commissioned, despite a

failure to reduce disease inequalities and disease prevalence.[42] They found that individual-level interventions are appealing to governments due to ideological and practical reasons and that political ideologies needed to be included more widely in research. Baum *et al* in a separate study undertook qualitative interviews of Australian primary care policy-makers, examining the primary care response to the social determinants of health.[75] They showed that primary care health workers saw their roles as advocates for individuals improving access to social services. Institutional scope and service demands acted as barriers to advocating for action on the wider determinants of health.

Maybin[76] undertook an ethnography of UK Department of Health and Social Care policy-makers. Powerful political actors, political priorities, relationships and trust determined what counted as knowledge in the policy process. Comparably, to our study, policy-makers used evidence symbolically to legitimise the decision-making process and funding.

## Strengths and limitations

To our knowledge, this is the first analysis of diabetes prevention policies in the UK using a political science perspective. Other studies have focused on either the policy process[69] or the research–policy gap,[73 74 77] but this is the first to examine both as part of the same social–political context to better reflect reality.

Our focus on a small number of interviewees means the findings are preliminary. There were no outlier opinions from the dataset which might have been ascertained from a larger sample size or if we had included policy-makers from different regions of the country. In addition, we may also have broadened the study to discuss the prevention of long-term conditions was more widely. A further limitation of the study is that most of the data was collected before COVID-19 and while we picked up on the way the pandemic amplified existing issues, we did not explicitly explore this in data collection/analysis.

## CONCLUSION

This qualitative study of policy-maker perspectives on national diabetes prevention policies and their implementation has allowed us to better understand how policy decisions are made and why current policies are constructed in their present form. Current strategies align with a neoliberal high-risk disease prevention strategy, which focuses on individuals to take responsibility for their lifestyles and make changes to prevent disease. Although these policies work for some individuals to reduce their risk, continuing to perpetuate individual-level disease prevention strategies may widen health inequalities. Diabetes prevention policy sits at the intersection of public health and primary care. The advent of ICSs and primary care networks may offer an opportunity to focus on population-level drivers of disease as well as open up a new research agenda to bridge evidence–policy gaps.

**Contributors** EB conducted the interviews, analysed the data and was the lead author who revised drafts of the paper. TG, SS and CP provided supervision support during data analysis and assisted with revisions of the paper. EB acts as a guarantor for the study.

**Funding** EB is funded by an NIHR Doctoral Research Fellowship (DRF 2017-10-024).

**Competing interests** None declared.

**Patient and public involvement** Patients and/or the public were not involved in the design, or conduct, or reporting, or dissemination plans of this research.

**Patient consent for publication** Not applicable.

**Ethics approval** This paper forms one part of a wider study on pre-diabetes. Overall approval for the study was given by the University of Oxford Clinical Trials and Research Governance group, who confirmed that NHS ethics approval was not required for this stakeholder phase of the study because it did not involve or discuss NHS Patients (reference IRAS 242219). NHS Research Ethics Committee approval was received for other phases of the study involving patients and NHS staff, reference 18/LO/0479.

**Provenance and peer review** Not commissioned; externally peer reviewed.

**Data availability statement** The data that support the findings of this study are available on request from the corresponding author, EB. The data are not publicly available due to their containing information that could compromise the privacy of research participants. No datasets generated.

**ORCID iDs**
Eleanor Barry http://orcid.org/0000-0003-1655-6516
Trisha Greenhalgh http://orcid.org/0000-0003-2369-8088
Sara Shaw http://orcid.org/0000-0002-7014-4793

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
