## [Reviewer comments · BMJ Open]

ARTICLE DETAILS

TITLE (PROVISIONAL)	Explaining the UK's 'high-risk' approach to type 2 diabetes prevention- Findings from a qualitative interview study with policy makers in England.
AUTHORS	Barry, Eleanor; Greenhalgh, Trisha; Shaw, Sara; Papoutsis, Chrysanthi

VERSION 1 – REVIEW

REVIEWER	Suan Ee Ong Natl Univ Singapore
REVIEW RETURNED	30-Jul-2022

GENERAL COMMENTS	Reviewer comments: Barry et al 2022 for BMJ Open Page 6, Lines 21-50: • What was the informed consent process like? Please detail this (e.g., use of standardized forms, the basic content of the forms, how interviewees returned forms to the research team, whether verbal consent was audio recorded).• Please include some discussion on how the topic guide was developed and if it was informed by any frameworks, theories, or a literature review.• It would be helpful for readers and those interested in this work to have the topic guide available in the appendices.• Please mention if detailed notes/analytic memos were taken for each interview conducted, and if so, how they were used to inform analysis.• Please mention who transcribed the interview audio and whether post-transcription checks were conducted for accuracy and clarity.• Please mention of how data were securely stored and when/how data would be deleted upon completion of the study, if relevant.• Why were transcripts not returned to participants for comment and/or correction?• Please explain why participants did not provide feedback on the findings of this study. Page 6, Lines 41-43: • Why were only GP leads and the national health policymaker re-interviewed in 2021 in the context of COVID and gathering feedback on initial findings? It would be helpful to explain this decision. Page 9, Lines 3 onwards (Results section): • The results are very interesting, but would it be possible to share more details of interviewee perspectives that did not align with
--

	the majority of perspectives shared in this study? This would greatly enhance and enrich the section.  This section would be strengthened with the inclusion of more findings that explore how the COVID-19 pandemic had affected the policy process and decision-making. Page 15,  This section would benefit from some additional discussion of the COVID-19 context and how it has affected the health system in general. Page 16, Lines 48-49:  Please include a more detailed discussion of potential study limitations aside from the relatively low sample size.
--	---

REVIEWER	Samuel Stratton University of California, Los Angeles, Community Health Sciences
REVIEW RETURNED	14-Nov-2022

GENERAL COMMENTS	GENERAL COMMENT:  This is an interesting and informative manuscript. SPECIFIC COMMENTS:  What is the evidence to show failure of the current prevention program(s) as stated in the Abstract? There is no comparative data presented to show failure, improvement, or no effect. The study is based on 4 purposeful selected participants, but the remaining nine participants are recruited by "snowballing". This leads to major limitations of the study that should be noted, included selection bias, inability to determine sampling error, and bias due to motivation of participants to be included in the study. Because the study is based upon purposeful and snowball sampling, the participants cannot be stated to represent the overall policy-maker group in the UK (participants were not randomly selected) and therefore, the study results cannot be generalized beyond the 12 study participants. It does not appear that all participants in the study are actually policy-makers (target group). The relationship and policy-making authorities of each participant should be stated to assure the study group represents a policy-maker group. The last two sentences of the Conclusion seem to come from outside the study. Based upon the data presented, the integrated care systems are not the focus of the study and not well established as a solution to the problems identified in the paper.
--

REVIEWER	JM Lowe The University of Newcastle, University of Newcastle
REVIEW RETURNED	20-Nov-2022

GENERAL COMMENTS	Thank you for your fascinating article. I have a couple of minor suggestions to improve clarity. In the abstract under Data Collection lines 30 -32 the sentence does not make sense. On page 16 lines 49-50 I suggest "perspective" conveys the meaning better than "study"
--

VERSION 1 – AUTHOR RESPONSE

Reviewer Reference	Reviewer Comments	Author's Response
	Reviewer 1 comments	
R1.1	I enjoyed reading this paper and think that the research question is both relevant and interesting, thanks very much for writing it. My detailed comments are in the attached file.	We'd like to thank the reviewer for their comments. Many thanks for taking the time to highlight the additional information required in this manuscript. Apologies for its absence in the submitted manuscript, in an effort to reduce the word count some of this information was removed.
R1.2	Page 6, Lines 21-50: □ What was the informed consent process like? Please detail this (e.g., use of standardized forms, the basic content of the forms, how interviewees returned forms to the research team, whether verbal consent was audio recorded).	Written consent forms were completed by each participant and audio recording commenced once the forms had been completed (obtaining consent for recordings). This has been made explicit within the data collection section of the manuscript. We have uploaded an example consent form as a supplementary file. This paper forms one part of a wider study on pre-diabetes. Overall approval for the study was given by the University of Oxford Clinical Trials and Research Governance group, who confirmed that NHS ethics approval was not required for this stakeholder phase of the study (reference IRAS 242219). NHS Research Ethics Committee approval was received for other phases of the study involving patients and NHS staff reference 18/LO/0479. This explanation has been added to the Ethics section at the end of the manuscript.
R1.3	Please include some discussion on how the topic guide was developed and if it was informed by any frameworks, theories, or a literature review.	The development of topic guides was based on the current diabetes prevention policies (particularly NICE PH 38 and the NHS Diabetes Prevention Programme) referenced in the introduction section of the paper. We designed a topic guide was designed for each interview in an iterative way, building on the information gathered in previous interviews, to be able to ask questions and guide the conversation in a way relevant to each of the different stakeholders (sampled purposely to seek variation, as is usual in qualitative research), therefore leading to richer data. We have added additional text in the methods section to make this clear to readers.

		Theories were applied to the data set during data analysis (see methods and findings section, main paper).
R1.4	It would be helpful for readers and those interested in this work to have the topic guide available in the appendices.	We agree, this could be helpful for readers. An example has now been included as an appendix. Please note the topic guide was not given to participants but used by the first author to conduct the interviews.
R1.5	Please mention if detailed notes/analytic memos were taken for each interview conducted, and if so, how they were used to inform analysis.	Detailed notes and analytical memos were made after each interview and during the analysis process. We have added the following to clarify this in the paper “EB conducted the interviews, took field notes and maintained a reflective journal which assisted in the data analysis”
R1.6	Please mention who transcribed the interview audio and whether post-transcription checks were conducted for accuracy and clarity.	This information has been added to the data collection section of the manuscript. We have added the following sentence in the data collection section “Interviews were audio-recorded (with written consent), de-identified and then transcribed verbatim by a professional transcriber”
R1.7	Please mention of how data were securely stored and when/how data would be deleted upon completion of the study, if relevant.	All of the audio files were anonymised with a participant ID number. Audio files and transcripts were stored on a university computer on a password protected non-network drive by the participant ID number. The research data will be kept for up to three years after the end of the study’s data collection period, to allow time to analyse the data and write papers. Once these are completed audio data will be permanently deleted. Given the word limit we have not included this information in the paper but are happy to do so on advice from the editor
R1.8	Why were transcripts not returned to participants for comment and/or correction? Please explain why participants did not provide feedback on the findings of this study.	The lead author confirmed accuracy of each transcript against the audio file. The interviewees did not express willingness to review their transcripts, but three of them provided feedback on initial findings and themes. Findings from the study were also discussed and reviewed with the final participant who had experience as both a local and national policy maker within the NHS.
R1.10	Page 6, Lines 41-43: [ ] Why were only GP leads and the	Many thanks for pointing this out. No new information was ascertained by interviewing the three stakeholders again. Their feedback was that

	national health policymaker re-interviewed in 2021 in the context of COVID and gathering feedback on initial findings? It would be helpful to explain this decision.	covid-19 had amplified the existing funding imbalances, power dynamics and hierarchies around disease prevention. In addition, due to the pandemic most disease prevention initiatives had stopped and have only restarted in the last year or so. This information has been added to the data collection section.
R1.11	Page 9, Lines 3 onwards (Results section): [ ] The results are very interesting, but would it be possible to share more details of interviewee perspectives that did not align with the majority of perspectives shared in this study? This would greatly enhance and enrich the section. Page 15, [ ] This section would benefit from some additional discussion of the COVID-19 context and how it has affected the health system in general	Many thanks - this is a really important point. The dataset didn't contain any outlying opinions, with each stakeholder reflecting their organisational view point. We suspect that this may be due to the small sample size and that most stakeholders came from a similar area of the country. We have added this as a limitation to the study (see strengths and limitations section). Related to this point, Covid-19 was not the focus of this study it did come up in interviews as amplifying health inequalities and the importance of disease prevention. Hence, while it would be difficult to expand the findings or discussion to take Covid-19 context explicitly into account when the data was collected in a pre-covid time, we have added this as a limitation to the study.
R1.12	[ ] This section would be strengthened with the inclusion of more findings that explore how the COVID-19 pandemic had affected the policy process and decision-making.	As discussed previously many disease prevention initiatives stopped entirely during the covid pandemic, but we hope the results from this study can amplify the need for further collaborative working between public health and primary care moving forward.
R1.13	.	
R1.14	Page 16, Lines 48-49: [ ] Please include a more detailed discussion of potential study limitations aside from the relatively low sample size.	We have revised the limitations section as follows: "Our focus on a small number of interviewees means the findings are preliminary. There were no outlier opinions from the dataset which might have been ascertained from a larger sample size or if we had included policy makers from different regions of the country. In addition, we may also have broadened the study to discuss the prevention of long-term conditions was more widely. A further limitation of the study is that most of the data was collected pre-covid and while we picked up on the way the pandemic amplified existing issues, we did not explicitly explore this in

		data collection/analysis.”
	Reviewer 2	
R2.1	What is the evidence to show failure of the current prevention program(s) as stated in the Abstract? There is no comparative data presented to show failure, improvement, or no effect.	Many thanks for highlighting this sentence. We agree it is misleading and have removed it. We discuss the limitations to current diabetes prevention strategies in the introduction of the manuscript with reference to published evaluations.
R2.2	The study is based on 4 purposeful selected participants, but the remaining nine participants are recruited by "snowballing". This leads to major limitations of the study that should be noted, included selection bias, inability to determine sampling error, and bias due to motivation of participants to be included in the study.	Many thanks to the reviewer for their comments. This qualitative study is not seeking statistical generalisation in a quantitative sense, but the themes and discussions from these elite interviews can be applied to other health care settings which take a similar approach to diabetes prevention. We accept that the sample size is a limitation of the study and have discussed this in the limitations section.
R2.3	Because the study is based upon purposeful and snowball sampling, the participants cannot be stated to represent the overall policy-maker group in the UK (participants were not randomly selected) and therefore, the study results cannot be generalized beyond the 12 study participants.	See comment above.
R2.4	It does not appear that all participants in the study are actually policy-makers (target group). The relationship and policy-making authorities of each participant should be stated to assure the study group represents a policy-maker group.	Table 1 outlines each participants policy role. In the UK GP clinicians play key policy roles, making decisions on health care policy and initiatives in their local areas. Giving further information than what is in the table would risk their anonymity, we have made this clear in the footer of table 1.
R2.5	The last two sentences of the Conclusion seem to come from outside the study. Based upon the data presented, the integrated care systems are not the focus of the study and not well established as a solution to the problems identified in the paper.	Many thanks for this observation, we have abbreviated the last sentence for clarity.
	Reviewer 3	

R3.1	Thank you for your fascinating article.	Many thanks to the reviewer for their comments.
R3.2	In the abstract under Data Collection lines 30 -32 the sentence does not make sense.	Many thanks, this sentence has been removed and agree it was misleading.
R3.3	On page 16 lines 49-50 I suggestive "perspective" conveys the meaning better than "study"	We have changed the wording.

VERSION 2 – REVIEW

REVIEWER	Samuel Stratton University of California, Los Angeles, Community Health Sciences
REVIEW RETURNED	18-Dec-2022

GENERAL COMMENTS	Thank you for addressing the concerns of the Reviewers in complete and appropriate ways.
--